# Ultrasonic Extraction of Polysaccharides from *Dendrobium officinale* Leaf: Kinetics, In Vitro Activities, and Characterization

**DOI:** 10.3390/foods13233737

**Published:** 2024-11-22

**Authors:** Xuerong Shi, Xuzhong Yang, Shaotong He, Ting Duan, Xin Liang, Shuzhen Ma, Jijun Gong

**Affiliations:** 1College of Food Science and Engineering, Central South University of Forestry and Technology, Changsha 410004, China; 2Hunan Key Laboratory of Forestry Edible Sources Safety and Processing, Hunan Key Laboratory of Processed Food for Special Medical Purpose, Changsha 410004, China

**Keywords:** *Dendrobium officinale* leaf, polysaccharide, ultrasonic extraction, in vitro bioactivity, characterization

## Abstract

This study explored the kinetics of ultrasonic extraction of polysaccharides (*DO*LP) from *Dendrobium officinale* leaf (*DO*L), evaluated the in vitro bioactivity of *DO*L extracts and *DO*LP, and characterized the *DO*LP. A kinetic model was developed based on Fick’s second law. A technique utilizing 400 W for 50 min was employed for the ultrasonic extraction of *DO*LP, with an optimal solid–liquid ratio established at 1:40 (g/mL). *DO*L extracts dried using different methods exhibited varying antioxidant activity and inhibitory effects against α-amylase and α-glucosidase. An in vitro study revealed that *DO*L extracts obtained through vacuum freeze drying demonstrated significantly stronger antioxidant activity, while those derived through microwave drying showed superior inhibitory effects against α-amylase and α-glucosidase compared to the other two drying methods. Furthermore, it was observed that the in vitro bioactivity of *DO*LP (purity: 74.07 ± 0.52%) was significantly lower than that of *DO*L extracts. Nevertheless, *DO*LP (5.0 mg/mL) demonstrated a scavenging ability reaching 64.86% of VC for DPPH radical and 67.14% of VC for ·OH radical, and the inhibition of *DO*LP (10 mg/mL) on α-amylase and α-glucosidase reached 58.40% and 38.28% of the acarbose, respectively. The findings revealed that *DO*LP are predominantly composed of mannose, glucose, galactose, and arabinose in a distinctive molar ratio of 89.00:16.33:4.78:1.

## 1. Introduction

*Dendrobium officinale* (*DO*), a perennial herb of the Orchidaceae and *Dendrobium* genus, is one of the most important Chinese medicinal herbs. It is also known as Tiepi Shihu in traditional Chinese medicine (TCM) and is extensively distributed worldwide [1]. The safety of *DO* has been well demonstrated, and due to its broad range of medical qualities, it is widely employed as an ingredient in medicines, nutraceuticals, and food products [2,3]. The most noticeable effects of *DO* include nourishing the lungs to alleviate coughing, tonifying the stomach, enhancing the production of bodily fluids, clearing heat, improving immunity, anti-aging, and lowering blood sugar levels [4]. The main components in *DO* are polysaccharides [5], phenolic compounds [6], and alkaloids [4]. *D. officinale* polysaccharides (*DO*P) have been identified as the main active component, with levels ranging from 10% to 50% in dried *DO* stems [5,7]. Polysaccharides are essential biological macromolecules that are present in a wide variety of organisms [7]. *DO*P exhibit antitumor [8], antioxidation [9], immunomodulatory [2,10], anti-inflammatory [11], and diabetes prevention properties [3]. Therefore, *DO*P extraction from *DO* is gaining popularity around the world.

*DO* stems have a long-standing history of medicinal applications; however, the leaves are typically discarded during production. Additionally, *DO* leaves (*DO*L) are usually peeled for aesthetic purposes [12]. As a result, the predominant focus of research on *DO* has been directed towards the stem, despite the fact that the leaf exhibits comparable yields and contains active chemicals. Liu et al. [13] found that polysaccharides isolated from the stem and leaf varied in structure and activity. Therefore, it is crucial to accurately determine the activity of polysaccharides in various sections of *DO*.

In recent years, research has increasingly demonstrated that the therapeutic properties of *DO*L derived from a variety of bioactive compounds. These include flavonoids with antioxidant [14,15], antidiabetic [16,17], and antihyperlipidemic properties [18]; alkaloids with anti-cancer properties [13,19]; and polysaccharides with antioxidant [5,15], anti-inflammatory [20,21,22], immune system modulation [22,23], and blood sugar level regulation activities [15,24]. It has been determined that *DO*L polysaccharides (*DO*LP) are the primary component, comprising 8.99–16.55% of the total content, while the range of total flavonoid content is from 7.66 to 9.50 mg/g dried weight [17].

Efficient extraction of polysaccharides is currently a popular research subject. The methods for extracting polysaccharides include water extraction and alcohol precipitation [25,26], fast solvent extraction [27], ultrasonic-assisted extraction [28,29], microwave-assisted extraction [30], ultrasonic-microwave-assisted extraction [31], enzymatic extraction [32,33], and freeze–thawing cold-pressing extraction [32]. Among the aforementioned methods, ultrasonication has gained significant traction in the extraction of bioactive polysaccharides due to its advantages in enhancing diffusivity, solubility, and transport of solute molecules. This method not only increases the extraction rate of polysaccharides but also reduces both the extraction time and solvent waste [28]. Recent research on *DO*LP has primarily focused on improving the purity and activity by enhancing the technique used in the extractions [13,17,20,34]. Although studies have been conducted on the optimization of ultrasound-assisted extraction of polysaccharides from the stems and leaves of *DO*, the optimized extraction conditions that yield a high quantity of polysaccharides were derived primarily by examining the relationship between extraction parameters and polysaccharide yield. However, the underlying principles governing polysaccharide dissolution, particularly their kinetics and thermodynamics under ultrasonic conditions, remain unclear. Recently, there have been reports indicating that different drying methods can influence the bioactivity of polysaccharides [15]. However, it remains uncertain how different drying methods affect the bioactivity of *DO*L extracts and *DO*LP. In the realm of contemporary polysaccharide research, purification has emerged as a crucial step [2,23,28]. Nevertheless, the impact of purifying *DO*L extracts on the bioactivity of *DO*LP also remains unclear.

The objective of this study was to establish the correlation between *DO*LP concentration and ultrasonic conditions (power and time) during the water extraction process, develop a kinetic model for *DO*LP extraction based on Fick’s second law, and determine key kinetic parameters, such as the rate constant, relative extraction remaining rate, half-life, and diffusion coefficient. The in vitro bioactivity of *DO*L extracts was assessed following various drying methods, including microwave drying (MD), hot air drying (HAD), and vacuum freeze drying (VFD). Furthermore, the impact of decolorization and deproteinization on the in vitro bioactivity of *DO*LP was investigated. Additionally, preliminary characterization of *DO*LP was conducted.

## 2. Materials and Methods

### 2.1. Materials and Pretreatment

*DO*L were purchased from Xishuangbanna, Yunnan, China. The leaves were subjected to drying in a hot air oven at 50 °C for 120 min, followed by grinding into powder and screening through a 60-mesh sieve.

### 2.2. Ultrasonic Water Extraction of DOLP

Three grams of *DO*L powder was placed in a beaker for the purpose of ultrasonic extraction. The frequency used for the ultrasonic processor (JY98-IIN, Ningbo Scientz Biotechnology Co., Ltd., Ningbo, China) was 21.00 KHz, with an ultrasonic interval of 1 s and a stopping time of 2.5 s. To maintain a constant temperature of 0 °C during ultrasonication, the ice-water method was employed, accompanied by a real-time temperature monitoring probe for effective oversight and control. The solid–liquid ratio (using ultra-pure water as the solvent) was tested at ratios of 1:20, 1:30, 1:40, 1:50, 1:60, and 1:70. The power (400 W) and duration (30 min) were referenced from the report by Chen et al. [11]. The suitable solid–liquid ratio was determined based on the yield of *DO*LP: *DO*LP yield (%) = G_1_/G_2_ × 100, where G_1_ is the content of *DO*LP (g) and G_2_ is the mass of *DO*L powder (g).

After extraction, the solution was centrifuged at 4000 rpm for 10 min. Then, a volume of 0.125 mL of the supernatant was diluted to a total volume of 25 mL in order to determine the concentration of *DO*LP. Ultrasonic power levels of 200 W, 250 W, 300 W, 350 W, and 400 W were tested along with ultrasonic time levels of 10 min, 20 min, 30 min, 40 min, 50 min, and 60 min at an optimal solid–liquid ratio. Kinetics analysis was conducted based on the concentration of *DO*LP (mg/mL) in the extraction solution at various ultrasonic power and time levels.

### 2.3. Development of Kinetic Model

The *DO*L particle is considered to be spherical, with R representing its radius (mm). In an ideal spherical model, the kinetic equation based on Fick’s second law can be expressed as Equation (1) [35]:(1)$$\frac{𝜕c}{𝜕t}=D_{s}\left(\frac{{𝜕}^{2}c}{𝜕r^{2}}+\frac{2}{r}\frac{𝜕c}{𝜕r}\right)$$
where t represents the ultrasonic time (min), c denotes the concentration of *DO*LP (mg/mL) at a distance of r (mm) from the spherical surface within the particle at the time of t (min), and Ds is the diffusion coefficient of *DO*LP (mm^2^/min).

Given f = c × r, then
(2)$$\frac{𝜕f}{𝜕t}=D_{s}\frac{{𝜕}^{2}f}{𝜕r^{2}}$$

When the boundary conditions are r = 0, f = 0, there is
(3)$$r=R,\;\left(\frac{𝜕C_{out}}{𝜕t}\right)\cdot V_{out}=-D_{s}S{\left(\frac{𝜕c}{𝜕r}\right)}_{r=R}$$
where C_out_ is the concentration of *DO*LP (mg/mL) in the solvent at time t, V_out_ denotes the solvent volume (mL), and S represents the contact area between *DO*L particles and the solvent (mm^2^).

Equation (3) can be transformed into Equation (4) using the Fourier transform approach [1]:(4)$$\left(C_{\infty }-C\right)/\left(C_{\infty }-C_{0}\right)=\left(6/\pi^{2}\right)\sum_{n=1}^{\infty }\left\{exp\left[-{\left(n\pi /R\right)}^{2}D_{s}t\right]\right\}$$
where C_0_ represents the initial concentration of *DO*LP in the solvent (mg/mL) and is the equilibrium concentration of *DO*LP in the solvent (mg/mL).

Due to the infinite order distribution equation of *DO*LP concentration, its high-order term tends to zero infinitely and can be ignored. Therefore, setting *n* = 1 in Equation (4) results in Equation (5) [36]:(5)$$\frac{C_{\infty }-C}{C_{\infty }-C_{0}}=\frac{6}{\pi^{2}}exp(-\pi^{2}Dst/R^{2})$$

In ultrasonic extraction, the diffusion coefficient consists of molecular diffusion and eddy diffusion caused by ultrasonic vibration. Therefore, *Ds* = *Do* + *Du*, resulting in Equation (6):(6)$$\frac{C_{\infty }-C}{C_{\infty }-C_{0}}=\frac{6}{\pi^{2}}exp(-\pi^{2}(Do+Du)\;t/R^{2})$$
where *Do* is the molecular diffusion coefficient of *DO*LP and *Du* represents the ultrasonic diffusion coefficient of *DO*LP. In the context of ultrasonic extraction, *Du* is significantly greater than *Do*, thus leading to an approximation that Ds ≈ Du.

Since *C*_0_ = 0, Equation (6) is transformed into Equation (7) by applying logarithms on both sides:(7)$$\ln\left(\frac{C_{\infty }-C_{0}}{C_{\infty }-C}\right)=\left(\pi^{2}D_{u}/R^{2}\right)t+\ln\left[\frac{\pi^{2}C_{\infty }}{6{(C}_{\infty }-C_{0})}\right]$$

Equation (8) is then derived from simplifying Equation (7):(8)$$\ln\left(\frac{C_{\infty }-C_{0}}{C_{\infty }-C}\right)=\mathrm{kt}+\ln\left[\frac{\pi^{2}C_{\infty }}{6{(C}_{\infty }-C_{0})}\right]$$
where k is the apparent rate constant (k = $\pi^{2}D_{u}/R^{2}$).

Equation (8) delineates the kinetic model utilized for guidance of ultrasonic water extraction of *DO*LP.

### 2.4. Analysis of the Content of DOLP, Total Flavone, Polyphenol, and Protein

The *DO*LP content was measured by the phenol–sulfuric acid method using Glc as a standard [37]. The total flavone content was determined at 510 nm using the colorimetric method, with aluminum nitrate as the color-developing agent and rutin as the standard [38]. The polyphenol content was assessed at 760 nm using the Folin–Ciocalteu method [39]. Additionally, protein content was determined utilizing the Komas Brilliant Blue G-250 dyeing procedure [40].

### 2.5. Preparation of Samples of DOL Extracts and DOLP

#### 2.5.1. Samples of DOL Extracts

Twenty grams of *DO*L powder was placed in a beaker, ultra-pure water was used as the solvent, and the test was carried out at a material–liquid ratio of 1:40, with the ultrasonic power set at 400 W and the time set at 50 min for ultrasonic extraction. Following this, the extract was centrifuged at 4000 rpm for 15 min, and the supernatant was then concentrated by spinning at 50 °C. Subsequently, four times the volume of anhydrous ethanol was added to the concentrated extract, and alcohol precipitation was carried out for 24 h. Afterward, the alcohol precipitate was centrifuged at 4000 rpm for 15 min. Finally, the precipitate, referred to as *DO*L extracts, underwent a dilution process and was subsequently dried utilizing the following drying techniques.

MD: The *DO*L extracts were subjected to microwave vacuum drying using a WZD2S model (Nanjing Sanle Microwave Technology Development Co., Ltd., Nanjing, China), operating at a frequency of 2450 MHz and a power rating of 2 kW. The microwave power was set to 200 W, with each cycle lasting for 1 min followed by natural cooling for 2 min. The total duration of microwave exposure amounted to 45 min. HAD: The *DO*L extracts were carried out at 80 °C for 4 h using an electric constant temperature drying box (202-2AB, Tianjin Tester Instrument Co., Ltd., Tianjin, China). VFD: Initially, the *DO*L extracts were frozen at −20 °C for a period of 24 h. They were then transferred to a vacuum freeze-dryer (Scientz-N, Ningbo Scientz Biotechnology Co., Ltd., Ningbo, China), where they underwent drying at an ambient temperature of 25 °C, with the cold trap maintained at −50 °C and under a vacuum pressure of 10 Pa. This drying process continued for 37 h until a constant weight was achieved, indicating that the drying process had been successfully completed. Furthermore, the VFD method was also employed to dry *DO*LP.

#### 2.5.2. Sample of DOLP

The *DO*L extraction solution was concentrated to one-tenth of its original volume using a rotary evaporator. The concentrated solution was treated with four-fold anhydrous ethanol (*v*/*v*) for 24 h. Subsequently, the mixture was centrifuged at 4000 rpm for 10 min, and the resulting precipitates were vacuum freeze dried. The dried material was then transformed into a 15 mg/mL solution and deproteinized. This involved subjecting the sample solution to enzymatic hydrolysis with papain (1% *w*/*v*) at pH 7 and 50 °C for 2 h, followed by enzyme deactivation at 100 °C for 10 min. Additionally, the solution underwent deproteinization three times using Sevage reagent (chloroform:n-butanol = 4:1, *v*/*v*) [41]. Furthermore, the solution was concentrated and dialyzed in tap water for 24 h (MWCO: 3500 Da), followed by another round of dialysis in ultra-pure water for an additional 24 h [42]. Finally, polysaccharides known as *DO*LP were produced through vacuum freeze drying.

### 2.6. In Vitro Activity Assay of DOL Extracts and DOLP

#### 2.6.1. In Vitro Antioxidant Activity Assay

The antioxidant activities of *DO*L extracts and *DO*LP were assessed in vitro by measuring their ability to scavenge DPPH, ABTS, and hydroxyl (·OH) radical as well as their ferrous reducing power.

##### DPPH Radical Scavenging Activity Assay

The DPPH radical scavenging activity of *DO*L extracts and *DO*LP was assessed using the method outlined by Wang et al. [43], with some modifications. Specifically, 2 mL of 0.05, 0.50, and 5.00 mg/mL *DO*L extracts or *DO*LP along with an equal volume of 0.1 mmol/L DPPH solution (prepared with anhydrous ethanol and stored in the dark for 24 h) were individually added to a 10 mL colorimetric tube, thoroughly mixed, and allowed to react for 30 min at room temperature in the absence of light. The absorbance of the resulting mixture was measured at 517 nm and recorded as A_1_. The absorbance values for the control group without *DO*L extracts or *DO*LP (A_0_) and for the background without DPPH radical (A_2_) were also recorded. Ascorbic acid (VC) served as a positive control in subsequent antioxidant experiments.

The scavenging activity was determined using the following formula: DPPH radical scavenging activity (%) = (A_0_ − A_1_ + A_2_)/A_0_ × 100%.

##### ABTS Radical Scavenging Activity Assay

The ABTS radical scavenging activity of *DO*L extracts and *DO*LP was tested following the method described by Wang et al. [44], with minor adjustments. Briefly, a mixture of ABTS solution (7 mmol/L) and potassium persulfate solution (1.4 mmol/L) at a 1:1 (*v*/*v*) ratio was incubated in the dark at room temperature for 12 h to generate the ABTS radical. The ABTS working solution was prepared by diluting the ABTS radical solution with ethanol until its absorbance at 734 nm reached 0.70 ± 0.02. Subsequently, 0.4 mL of *DO*L extracts or *DO*LP at concentrations of 0.05, 0.50, and 5.00 mg/mL along with 4 mL of ABTS working solution were added separately to a 5.0 mL colorimetric tube, mixed thoroughly, and allowed to react at room temperature for 10 min. The absorbance was measured at 734 nm and recorded as A_1_. The absorbance values for the control group without *DO*L extracts or *DO*LP as well as the background without ABTS working solution were recorded as A_0_, and A_2_, respectively.

The scavenging activity was calculated using the following formula: ABTS radical scavenging activity (%) = (A_0_ − A_1_ + A_2_)/A_0_ × 100%.

##### OH Radical Scavenging Activity Assay

The OH scavenging activity of *DO*L extracts and *DO*LP was determined using the method described by Wang et al. [44], with some alterations. To summarize, 1 mL of 9 mmol/L ferrous sulfate and 1 mL of 9 mmol/L salicylic acid–ethanol solution (50% ethanol) were added individually to a tube. In addition, 1 mL of 0.05, 0.50, and 5.00 mg/mL *DO*L extracts or *DO*LP, 1 mL of VC solution, and 1 mL of 9 mmol/L H_2_O_2_ were added individually in different tubes and bathed in water at 37 °C for 30 min, and the absorbance was measured at 510 nm and recorded as A_1_. The absorbances of the control group with *DO*L extracts or *DO*LP solution replaced by deionized water and with H_2_O_2_ replaced by deionized water were measured as A_0_ and A_2_, respectively.

The scavenging activity was determined using the following formula: The ·OH scavenging activity (%) = (A_0_ − A_1_ + A_2_)/A_0_ × 100%.

##### Ferrous Reducing Power Assay

The ferrous reducing power of *DO*L extracts and *DO*LP was measured using the method reported by Zhu et al. [45], with some adjustments. In brief, 1 mL of 0.05, 0.5, and 5 mg/mL *DO*L extracts or *DO*LP was individually mixed with 2.5 mL of phosphate buffer (0.2 mol/L, pH = 6.6) and 2.5 mL of potassium ferricyanide (1 g/100 mL) at 50 °C for 20 min. The mixture was then treated with 2.5 mL of trichloroacetic acid (10 g/100 mL) and centrifuged at 3000 rpm for 10 min. After centrifugation, the supernatant (2.5 mL) was mixed with an equal volume of deionized water and then combined with 0.5 mL of ferric chloride (0.1 g/100 mL). After 10 min, the absorbance was measured at 700 nm and recorded as A_x_. The absorbance of the mixture with potassium ferricyanide replaced by deionized water was recorded as A_x0_.

The ferrous reducing power was determined using the following formula: ∆A_700_ = A_x_ − A_x0_.

#### 2.6.2. In Vitro α-Amylase Inhibitory Activity Assay

The α-amylase inhibitory activity of *DO*L extracts and *DO*LP was measured using the method described by Mittal et al. [46], with a few alterations. In test tubes, 0.5 mL of 0.5, 2.5, 5.0, 7.5, and 10.0 mg/mL *DO*L extracts or *DO*LP and 0.5 mL of acarbose solution (positive control) were individually mixed with 0.5 mL of porcine pancreatic α-amylase solution (0.5 U/mL, prepared with 0.1 mol/L, pH 6.9 phosphate buffer). Both solutions were incubated at 37 °C for 10 min before adding 0.5 mL of 1.0% soluble starch, mixed completely, and incubated for another 10 min. To halt the reaction, 1 mL of 3,5-dinitrosalicylic acid reagent was added and heated in a boiling water bath for 5 min. After cooling, each sample was diluted with ultra-pure water to a total volume of 20 mL. The reaction mixture was measured at 540 nm. The absorbance of the mixture containing *DO*L extracts or *DO*LP and α-amylase was recorded as A_1_. The absorbance of the mixture in which α-amylase was replaced by phosphate buffer was recorded as A_2_, while the absorbance of the mixture with *DO*L extracts or *DO*LP solution replaced by phosphate buffer was recorded as A_0_.

The α-amylase inhibitory activity of the samples was calculated using the following formula: Inhibition (%) = (A_0_ − A_1_ + A_2_)/A_0_ × 100.

#### 2.6.3. In Vitro α-Glucosidase Inhibitory Activity Assay

The α-glucosidase inhibitory activity of *DO*L extracts and *DO*LP was measured using the method reported by Wang et al. [47], with some changes. For the experiment, 100 μL of 0.5, 2.5, 5.0, 7.5, and 10.0 mg/mL *DO*L extracts or *DO*LP and 100 μL of acarbose solution were individually combined with 100 μL of α-glucosidase solution (1 U/mL) and incubated at 37 °C for 10 min. Subsequently, 100 μL of p-nitrophenyl-α-D-glucopyranoside (5.0 mmol/L) was added to the mixture and incubated for an additional 15 min at 37 °C. The reaction was terminated by adding 5 mL of sodium carbonate solution (0.2 mol/L). After a 10 min incubation at room temperature, the absorbance was measured at 405 nm. The absorbance of the mixture with *DO*L extracts or *DO*LP and α-glucosidase was recorded as A_s_. The absorbance of the mixture with α-glucosidase replaced by phosphate buffer was recorded as A_b_. The absorbance of the mixture with *DO*L extracts or *DO*LP solution replaced by ultra-pure water was recorded as A_0_.

The α-glucosidase inhibitory activity of the samples was calculated using the following formula: Inhibition (%) = [1 − (A_s_ − A_b_)/A_0_] × 100.

### 2.7. UV and FT−IR Spectrum Analysis

The ultraviolet absorbance of the *DO*LP solution (1 mg/mL) was measured by a spectrophotometer (UV-2600, Shimadzu, Kyoto, Japan) with a spectrum range of 190 nm to 400 nm.

The Fourier transform infrared (FT-IR) spectrum of *DO*LP was acquired using the KBr-pellets method on an FT−IR spectrometer (Thermo Nicolet IS5, Bruker, Billerica, MA, USA) within the range of 400–4000 cm^−1^.

### 2.8. Monosaccharide Composition Analysis

Ion chromatography was employed to analyze the monosaccharide composition of *DO*LP. Briefly, 5 mg of the sample was accurately weighed and placed in an ampoule containing 2 mL of 3 M trichloroacetic acid, which was then hydrolyzed at 120 °C for 3 h. The solution resulting from the acid hydrolysis was carefully transferred to a tube and dried using nitrogen gas. Subsequently, 5 mL of deionized water was added and vortexed gently. Then, 950 µL of deionized water was mixed with 50 µL of the sample solution. The mixture underwent centrifugation at 12,000 rpm for 5 min. The supernatant was analyzed using an ICS5000 (ThermoFisher, Waltham, MA, USA) system with a Dionex Carbopac^TM^ PA20 column (3 mm × 150 mm). The mobile phase consists of A: H_2_O, B: 250 mM NaOH, and C: 500 mM NaOH and 50 mM NaAc. The elution gradient proceeded as follows: 0 min, A/B/C (98:2:0, *v*/*v*); 23 min, A/B/C (98:2:0, *v*/*v*); 23.1 min, A/B/C (80:20:0, *v*/*v*); 33 min, A/B/C (80:20:0, *v*/*v*); 33.1 min, A/B/C (80:0:20, *v*/*v*); 46 min, A/B/C (80:0:20, *v*/*v*); 46.1 min, A/B/C (20:0:80, *v*/*v*); 66 min, A/B/C (20:0:80, *v*/*v*); 66.1 min, A/B/C (98:2:0, *v*/*v*); 80 min, A/B/C (98:2:0, *v*/*v*). The flow rate was maintained at a constant value of 0.3 mL/min, while the column temperature remained steady at 30 °C. A 254 nm diode array detector was utilized to monitor the analysis process. The standards were treated simultaneously with the samples.

### 2.9. Statistical Analysis

Data analysis and plotting were conducted using Origin 8.5 software and Microsoft Excel 2010. The experiments were designed with three replications, and the data were presented as mean ± standard deviation. The analysis of variance (ANOVA), followed by one-way Duncan’s test, were conducted using SPSS 27 software to evaluate the significant differences (*p* < 0.05) among treatment means.

## 3. Results

### 3.1. Kinetics of Ultrasonic Extraction of DOLP

#### 3.1.1. The Effect of Solid–Liquid Ratio on the DOLP Yield

Figure 1A depicts the impact of the solid–liquid ratio on *DO*LP yield. As shown in Figure 1A, increasing the solid–liquid ratio from 1:20 (g/mL) to 1:40 (g/mL) significantly enhanced *DO*LP yield (*p* < 0.05). However, further increases in the ratio led to a significant decrease in *DO*LP yield (*p* < 0.05). The maximum value was observed at a solid–liquid ratio of 1:40 (g/mL). Previous studies have also indicated an optimal solid–liquid ratio in polysaccharide extraction [48,49]. It is likely that exceeding a certain threshold in solution volume diminishes intermolecular interactions. Additionally, an excessive solid–liquid ratio hinders the medium′s ability to efficiently absorb ultrasonic energy, resulting in increased energy consumption [11]. Therefore, a solid–liquid ratio of 1:40 (g/mL) was found to be optimal for *DO*LP extraction.

#### 3.1.2. The Effect of Ultrasonic Power and Time on the DOLP Concentration

Table 1 shows the variations in *DO*LP concentration under different ultrasonic power and time settings while maintaining a fixed solid–liquid ratio of 1:40 (g/mL). The ultrasonic time was found to be a significant factor impacting *DO*LP yield, with an increase observed as the ultrasonic time was prolonged. Another important factor influencing polysaccharide yield was ultrasonic power, which affected the solubility of the target in the solvent and generated heat [11,50]. Polysaccharides are commonly present in plant cell walls. The production of massive cavitation bubbles by ultrasonic waves is responsible for the increased polysaccharide output. These bubbles burst rapidly, generating high shear and microjets that facilitate cell wall disintegration and solvent penetration [28,51]. The increase in *DO*LP concentration became less significant when the ultrasonic power exceeded 350 W. This can be attributed to excessive ultrasonic power affecting the transmission of energy into the medium by increasing bubble formation during cavitation [28,52]. When the *DO*LP concentration remained constant or changed minimally with increasing ultrasonic power and time, extraction equilibrium was reached. Table 1 shows that equilibrium (C_∞_) was reached at 350 W for 60 min or 400 W for 50 min (*p* > 0.05). To enhance extraction efficiency, employing a strategy of using 400 W for 50 min is preferred.

#### 3.1.3. Rate Constant Determination

Based on the data presented in Table 1, Figure 1B illustrates a plot depicting the linear relationship between ln[C_∞_/(C_∞_ − C)] and ultrasonic time (t) at various ultrasonic power levels.

Table 2 displays the linear regression equations, correlation coefficients (R^2^), and rate constants derived from the linear regression curves shown in Figure 1B for ln[C_∞_/(C_∞_ − C)] and ultrasonic time at various ultrasonic power levels. The R^2^ values of the linear regression equations ranged from 0.9520 to 0.9775, indicating a strong correspondence between the experimental results and the estimated values of the kinetic model. This suggests that ultrasonic water extraction of *DO*LP follows Fick’s second law and that the kinetic model is suitable for predicting *DO*LP dissolution behavior [35]. Furthermore, it was observed that the rate constant (k) significantly increased with enhanced ultrasonic power, implying a corresponding increase in the rate of *DO*LP dissolution.

#### 3.1.4. Relative Extraction Remaining Rate Determination

The relative extraction remaining rate is defined as the ratio of *DO*LP that have not yet been dissolved from the *DO*L particle to the amount dissolved into the extraction solvent at equilibrium. It is reasonable to assume that the initial concentration (C_0_) is zero. Under these conditions, the relative extraction remaining rate can be expressed as y = [(C_∞_ − C)/C_∞_], and Equation (8) can be modified to y = (6/$\pi^{2}$)exp (−k′t). The data in Table 1 were utilized to create Figure 1C, with ultrasonic extraction time on the horizontal axis and (C_∞_ − C)/C_∞_ on the vertical axis. Additionally, regression equations were fitted and are presented in Table 3.

Table 3 shows that, at a solid–liquid ratio of 1:40 (g/mL) and different ultrasonic power levels during *DO*LP extraction, the correlation coefficient (R^2^) of the fitted exponential equations between relative extraction rate Y = (C_∞_ − C)/C_∞_ and extraction time ranged from 0.9780 to 0.9991, indicating good curve-fitting accuracy consistent with an exponential model during ultrasonic extraction procedures. Figure 1C illustrates an exponential decline in the relative extraction remaining rate of *DO*LP as the ultrasonic extraction time increased. It is suggested that prolonging ultrasonic extraction time within a specific range benefits dissolution of *DO*LP from the *DO*L particle. Our findings are partially consistent with those reported by Zhang et al. [53], who investigated the extraction kinetic model of polysaccharides from Chinese chive.

#### 3.1.5. Half-Life Determination

The half-life is defined as the time required for polysaccharides to reach half of their equilibrium concentration. In Figure 1D, the relationship between t_1/2_ = ln2/k and ultrasonic extraction power is illustrated, with a regression equation of t_1/2_ = −0.0253P + 21.6704 (R^2^ = 0.9589). This equation can be utilized to determine the duration needed to achieve half of the equilibrium concentration. As ultrasonic extraction power increased, the half-life decreased, indicating a negative correlation between the two variables. A shorter half-life corresponds to a faster extraction rate, making it a reliable predictor of ultrasonic extraction efficiency. This also explains why increasing ultrasonic extraction power helps accelerate the *DO*LP extraction process [35].

#### 3.1.6. Diffusion Coefficient (Du) Determination

The apparent rate constant (k = $\pi^{2}$*Du*/R^2^), which measures the speed at which the extract diffuses within the medium, can be utilized to calculate *Du*. Equation (8) demonstrates that the apparent rate constant is dependent on *Du* and the particle radius [37]. Figure 1E was constructed by plotting 10^4^*Du* and the ultrasonic extraction power, assuming that the *DO*L particle was spherical and of fixed size (R = 0.125 mm). It is evident from Figure 1E that 10^4^*Du* increased as the ultrasonic extraction power intensified. The regression equation was found to be 10^4^*Du* = 0.4066exp (0.0018P) (R^2^ = 0.9685). However, R^2^ was slightly lower, possibly due to fluctuations in *DO*L particle size caused by ultrasound-induced aggregation during the actual extraction process under assumed conditions. These fluctuations impacted the accuracy of the fitting equation.

### 3.2. Samples of DOL Extracts and DOLP as Well as Changes After Purification

Figure 1F presents images of *DO*L extracts prepared using various drying methods as well as *DO*LP by VFD. As shown in Figure 1(F(a–c)), the color of *DO*L extracts by VFD was notably impressive compared to the other two samples. Meanwhile, the *DO*LP exhibited a milk-white color and fluffy condition (Figure 1F(d)).

Table 4 illustrates the variations in the purity of *DO*LP and the content of three chemicals following purification of *DO*L extracts. The *DO*LP content in the *DO*L extracts was significantly higher than other components, and there was also an impressive amount of polyphenol remaining, even after purification. The purification process led to a significant increase in the purity of *DO*LP (*p* < 0.05) while also significantly reducing the content of total flavone, polyphenol, and protein (*p* < 0.05).

### 3.3. In Vitro Antioxidant Activity of DOL Extracts and DOLP

Normally, all organisms, including humans, produce a certain amount of reactive oxygen species (ROS), which is beneficial to human health. However, an overabundance of ROS can inflict damage on essential macromolecules, thereby increasing the likelihood of disease [43]. Figure 2A–D illustrate the in vitro antioxidant activity of *DO*L extracts and *DO*LP. The *DO*L extracts and *DO*LP exhibited dose-dependent scavenging activity against DPPH, ABTS, OH radicals, and ferrous reducing power. Recent research has also indicated that polysaccharides demonstrate antioxidant activity in a dose-dependent manner [54,55,56]. Figure 2A–D show that the in vitro activity of VC was significantly greater than that of *DO*L extracts and *DO*LP at all three doses (*p* < 0.05). However, the latter also demonstrated remarkable ability except for lower ferrous reducing power. Among the three drying procedures, the VFD-derived *DO*L extracts showed an advantage in antioxidant activities except for ferrous reducing power, particularly at a concentration of 5.0 mg/mL, compared to the other two drying methods (*p* < 0.05). A previous study has shown that different drying methods for *DO*L extracts have varying effects on the protection of supercoiled DNA from oxidative damage [15]. At 5.0 mg/mL, the *DO*L extracts obtained by VFD exhibited significant scavenging activity against DPPH, ABTS, and ·OH radicals as well as ferrous reducing power, with values reaching 97.85%, 93.93%, 85.21%, and 23.79% of VC, respectively. Interestingly, *DO*LP revealed significantly lower in vitro antioxidant activity than *DO*L extracts (*p* < 0.05). This can be attributed to the purification process, which involves the removal of pigments, flavones, peptides, proteins, and polyphenols, leading to a decrease in antioxidant activity of polysaccharides [57,58,59]. However, at 5.0 mg/mL, *DO*LP demonstrated a scavenging ability reaching 64.86% of VC for DPPH radical and 67.14% of VC for OH radical. Hence, it is pertinent to conclude that *DO*LP also demonstrate satisfactory antioxidant activity in this study.

Polysaccharides seldom exist in isolation; instead, they typically conjugate with protein, lipid, and other compounds, and these polysaccharide conjugates may exhibit behavior characteristic of a unified entity during the process of isolation [60,61]. Cereal polysaccharides, for example, are known to link to phenolic compounds [61,62]. Tea polysaccharides are primarily classified as glycoconjugates, wherein proteins carry a carbohydrate chain that is covalently linked to a polypeptide backbone [61,63]. Conjugates of polysaccharides and polyphenols can be formed through hydrogen bonding or hydrophobic interactions. In contrast, conjugates between polysaccharides and proteins may arise due to the presence of hydrophobic cavities and crevices [64]. Previous studies have revealed that the antioxidant abilities of crude polysaccharides from tea leaves are reliant on tea polyphenols [61]. Lin et al. [65] proposed that the ability of hydroxyl groups in polysaccharides to suppress superoxide and hydroxyl radicals was inadequate due to the lack of a phenolic-type structure. Additionally, Wang et al. [59] discovered that the flavonoid fraction derived from *Lycium barbarum* L. exhibited the highest efficacy in scavenging DPPH and ABTS free radicals as well as in chelating metal ions and demonstrating reducing power. In contrast, the zeaxanthin fraction and polysaccharides were most effective in scavenging superoxide anions and hydroxyl free radicals, respectively.

Thus, it is possible that numerous variables, such as molecular weight, galacturonic acid, and other chemical components found in polysaccharide fractions, influence polysaccharides’ antioxidant activity.

### 3.4. α-Amylase and α-Glucosidase Inhibitory Activity

To effectively manage and prevent type 2 diabetes, one efficient approach is to slow down the digestion of carbohydrates and the absorption of glucose in the intestine by inhibiting salivary/pancreatic α-amylases and membrane-bound brush-border α-glucosidases [46,66].

Figure 2E,F, and Table 5 present the inhibition rates against α-amylase and α-glucosidase as well as the IC_50_ values for *DO*L extracts and *DO*LP, respectively. Both *DO*L extracts and *DO*LP demonstrated a dose-dependent impact on inhibiting α-amylase and α-glucosidase. At equivalent concentrations, the inhibitory activity of acarbose on both enzymes was superior to that of *DO*L extracts and *DO*LP (*p* < 0.05). Among the three drying methods tested, *DO*L extracts using MD showed the most effective inhibition on α-amylase and α-glucosidase at all concentrations except for 0.5 mg/mL (*p* < 0.05). Correspondingly, the IC_50_ values for α-amylase and α-glucosidase inhibition were significantly lower for the *DO*L extracts by MD compared to the other methods (Table 5). Drying methods affecting the inhibitory activities against α-amylase and α-glucosidase were also reported by Cai et al. [15], who evaluated the biological activities of *DO*L dried by different methods. At 10 mg/mL, it was observed that the inhibitory activity of the *DO*L extracts by MD reached 86.64% of acarbose for α-amylase and 74.59% of acarbose for α-glucosidase. Although the inhibition rate of *DO*LP for both enzymes was significantly lower (*p* < 0.05) compared with *DO*L extracts, it was noteworthy that, at 10 mg/mL, the inhibition of *DO*LP on α-amylase and α-glucosidase reached 58.40% and 38.28% of the acarbose, respectively. Overall, these findings suggest that both *DO*L extracts and *DO*LP have excellent inhibitory effects on α-amylase and α-glucosidase, indicating their potential use in hypoglycemic foods.

### 3.5. Pearson Correlation Analysis

The Pearson correlation coefficient was employed to examine the relationship between polysaccharide purity and in vitro activities as well as the association between activity indices [16]. Figure 3A,B demonstrate a strong correlation between *DO*LP purity and in vitro activities before and after purification, suggesting that *DO*LP were the most active component in the *DO*L extracts. Furthermore, there was also a significant positive correlation between in vitro activity indices.

### 3.6. UV and FT−IR Spectrum Analysis

The UV spectrum of *DO*LP is depicted in Figure 4A, revealing no significant characteristic absorptions at 260 nm and 280 nm. This indicates that the amount of nucleic acid and protein in *DO*LP is negligible.

The FT−IR spectrum of *DO*LP is shown in Figure 4B, with peaks at 3355 cm^−1^ and 2887 cm^−1^ representing the O-H and C-H bonds stretching vibration in the sugar ring, respectively [10,67]. The distinctive absorption at 1727 cm^−1^ can be attributed to the valence vibration of the C=O bond [10], while the absorption peak at 1642 cm^−1^ is related to the O-H bending vibrations [67]. The presence of uronic acid in *DO*LP is very low based on a comparison with four standards of uronic acid, contradicting the conjecture by Yang et al. [20]. The peak at 1373 cm^−1^ represents the symmetric C-H bending vibration of the methyl group [10], while the peaks at 1243 cm^−1^ and 1027 cm^−1^ indicate the stretching vibrations of C-O-C and the presence of a pyranose-type sugar, respectively [7]. The peaks at 809 cm^−1^ and 873 cm^−1^ in the spectrum denote the existence of a β-configuration of the sugar units in *DO*LP [7].

### 3.7. Monosaccharide Composition Analysis

The monosaccharide composition of *DO*LP was analyzed using an ion chromatography method. In Figure 4C, the ion chromatography profile of fifteen mixed standards, Fuc, GalN, Rha, Ara, GlcN, Gal, Glc, Xyl, Man, Fru, Rib, GalA, GulA, GlcA, and ManA, is displayed. By comparing the retention time of the standards and utilizing the standard curve for each monosaccharide, the monosaccharide composition of the sample was determined, and the molar ratio of monosaccharides in the polysaccharides was calculated.

As shown in Figure 4D, *DO*LP are composed of mannose, glucose, galactose, and arabinose in a molar ratio of 89.00:16.33:4.78:1. This differs with Zhong et al.’s findings [22], which identified the two types of polysaccharides from the leaves of *Dendrobium officinale* as DLP-1 and DLP-2 and found that DLP-1 was mainly composed of mannose (71.69%) and glucose (22.89%), whereas DLP-2 was constituted by rhamnose (35.05%), arabinose (24.12%), and galactose (25.65%) based on different chromatographic separations. Our results contradict those of Li et al. [34], who reported a molar ratio of 62.34:3.48:1.57:1 for the composition of mannose, glucose, galactose, and arabinose in *DO*LP. Furthermore, our findings are also inconsistent with those presented by Xiong et al. [17], who found that LDOP-A was composed of glucose, mannose, galactose, and arabinose with a molar ratio of 5.3:3.5:1.0:0.2. The aforementioned discrepancies may be attributed to the influence of geographical factors and variations.

## 4. Conclusions

For the first time, a kinetic model and its corresponding parameters for the ultrasonic extraction of *DO*LP utilizing water as the solvent were developed. The experimental results are in good agreement with the kinetic model. The kinetic model and derived parameters can serve as a valuable reference for *DO*LP extraction in production. This research meticulously investigated the effect of different drying methods on the antioxidant and α-amylase/α-glucosidase inhibitory activity of *DO*L extracts. The results demonstrated that *DO*L extracts by VFD exhibited stronger antioxidant activity, while those obtained via MD showed higher inhibition against α-amylase and α-glucosidase compared to the other two drying techniques. Furthermore, it was found that the purification procedure had a significant impact on the in vitro bioactivity of *DO*L extracts. This suggests that, in addition to polysaccharides, flavonoids, and polyphenols, proteins may also play an important role in the bioactivity of the *DO*L extracts. It is noteworthy that both the *DO*L extracts and *DO*LP demonstrated satisfactory bioactivity in this study. Additionally, our research revealed an intriguing discovery regarding the unique molar ratio of mannose, glucose, galactose, and arabinose in the composition of *DO*LP. As our measurements for characterizing *DO*LP in this study are preliminary at best, further investigation into its structure and form of conjugates is warranted. The findings imply that *DO*L extracts and *DO*LP have a promising future as an adjuvant and dietary supplement with antioxidant and hypoglycemic effects, indicating their potential for a wide range of utilizations.

## Figures and Tables

**Figure 1 foods-13-03737-f001:**
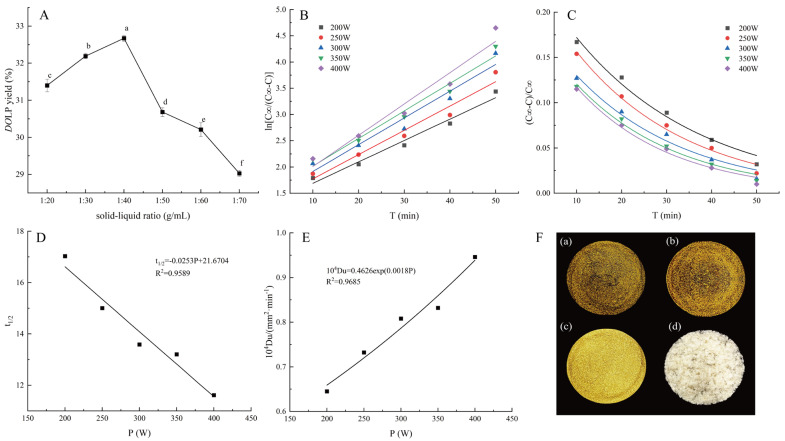
*DO*LP yield under different solid−liquid ratios (**A**); Relationship between ln[C_∞_/(C_∞_ − C)] and ultrasonic extraction time at different ultrasonic power levels (**B**); Relationship between (C_∞_ − C)/C_∞_ and ultrasonic time at different ultrasonic power levels (**C**); Relationship between t_1/2_ and ultrasonic power (**D**); Relationship between 10^4^*D_u_* and ultrasonic power (**E**); *DO*L extracts by MD (**F**(**a**)), HAD (**F**(**b**)), VFD (**F**(**c**)), and *DO*LP by VFD (**F**(**d**)). Different lowercase letters in Figure 1A indicate significant differences (*p* < 0.05).

**Figure 2 foods-13-03737-f002:**
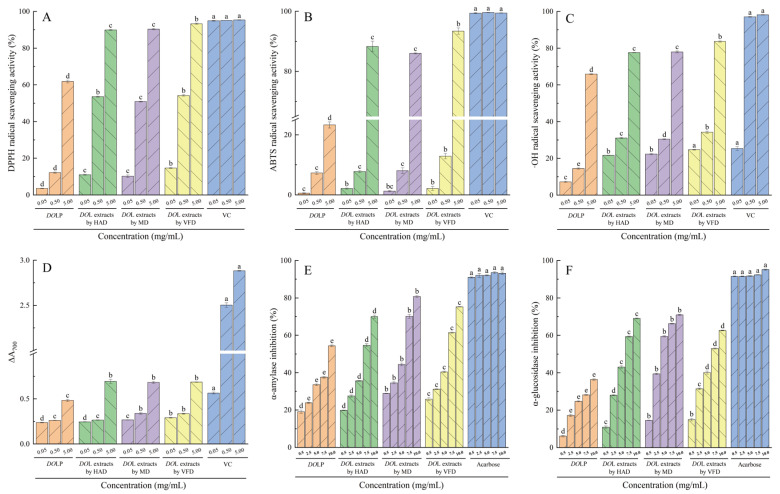
In vitro activity of *DO*LP. DPPH radical scavenging activity (**A**); ABTS radical scavenging activity (**B**); Hydroxyl radical-scavenging activity (**C**); Ferrous reducing power (**D**); α-amylase inhibitory activity (**E**); α-glucosidase inhibitory activity (**F**). Different lowercase letters indicate significant differences (*p* < 0.05) at the same concentration.

**Figure 3 foods-13-03737-f003:**
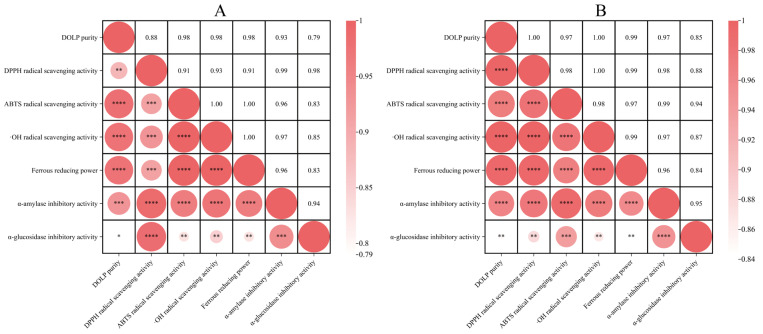
Heat maps of Pearson correlation analysis: before (**A**) and after purification (**B**); * *p* < 0.05, ** *p* < 0.01, *** *p* < 0.001, **** *p* < 0.0001.

**Figure 4 foods-13-03737-f004:**
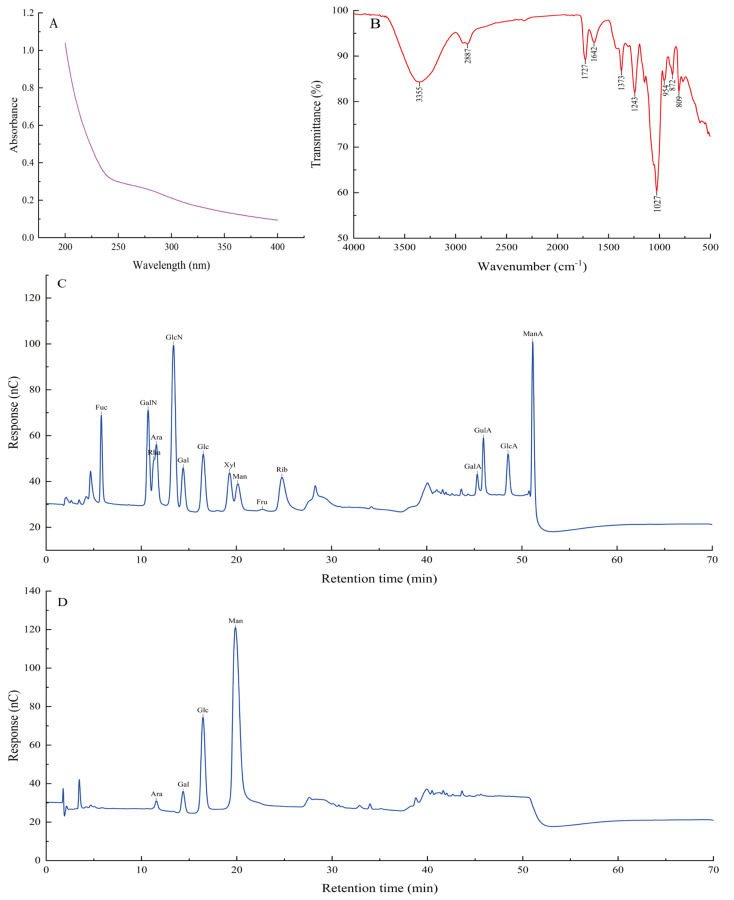
UV spectrum of *DO*LP (**A**); FT−IR spectrum of *DO*LP (**B**); Ion chromatography to determine the monosaccharide compositions of monosaccharide standards (**C**); *DO*LP (**D**).

**Table 1 foods-13-03737-t001:** *DO*LP concentration (mg/mL) in the extraction solution at different ultrasonic power and time levels.

Ultrasonic Power/W					
Ultrasonic Time/min	200	250	300	350	400
10	7.552 ± 0.074 q	7.874 ± 0.081 p	8.180 ± 0.121 o	8.424 ± 0.058 lm	8.501 ± 0.069 kl
20	7.904 ± 0.061 p	8.310 ± 0.057 mn	8.524 ± 0.061 kl	8.769 ± 0.053 j	8.891 ± 0.040 i
30	8.256 ± 0.057 no	8.608 ± 0.113 k	8.753 ± 0.105 j	9.052 ± 0.040 gh	9.144 ± 0.060 fg
40	8.528 ± 0.047 kl	8.838 ± 0.087 ij	9.021 ± 0.048 h	9.243 ± 0.058 def	9.342 ± 0.070 cd
50	8.776 ± 0.046 ij	9.098 ± 0.128 gh	9.220 ± 0.074 ef	9.419 ± 0.069 bc	9.519 ± 0.013 ab
60	9.067 ± 0.080 gh	9.304 ± 0.105 cde	9.365 ± 0.166 c	9.549 ± 0.115 a	9.610 ± 0.117 a

Different lowercase letters indicate significant differences at different ultrasonic power and time levels (*p* < 0.05).

**Table 2 foods-13-03737-t002:** Regression results for ln[C_∞_/(C_∞_ − C)] and extraction time at different ultrasonic extraction power levels.

Ultrasonic Powers/W	Regression Equation	R^2^	k/min^−1^
200	ln[C_∞_/(C_∞_ − C)] = 0.0407t + 1.2826	0.9757	0.0407
250	ln[C_∞_/(C_∞_ − C)] = 0.0462t + 1.3125	0.9629	0.0462
300	ln[C_∞_/(C_∞_ − C)] = 0.0510t + 1.4053	0.9520	0.0510
350	ln[C_∞_/(C_∞_ − C)] = 0.0525t + 1.4920	0.9705	0.0525
400	ln[C_∞_/(C_∞_ − C)] = 0.0597t + 1.4113	0.9775	0.0597

**Table 3 foods-13-03737-t003:** Regression results for (C_∞_ − C)/C_∞_ and ultrasonic extraction time under different ultrasonic extraction power levels.

Ultrasonic Powers/W	Regression Equation	R^2^
200	Y = 0.2452exp(−0.0355t)	0.9991
250	Y = 0.2327exp(−0.0398t)	0.9873
300	Y = 0.1956exp(−0.0407t)	0.9780
350	Y = 0.1875exp(−0.0441t)	0.9890
400	Y = 0.1880exp(−0.0476t)	0.9883

**Table 4 foods-13-03737-t004:** Changes after purification of *DO*L extracts.

	*DO*LP Purity/%	Total Flavone Content (g/100 g)	Polyphenol Content (g/100 g)	Protein Content (mg/g)
Unpurified (vacuum freeze drying)	44.90 ± 0.45 b	0.97 ± 0.03 a	18.34 ± 0.82 a	3.90 ± 0.04 a
Purifying treatments: Ethanol + papain hydrolysis combined with Sevage reagent + dialysis (vacuum freeze drying)	74.07 ± 0.52 a	0.27 ± 0.05 b	8.22 ± 0.14 b	0.37 ± 0.01 b

Different lowercase letters indicate significant differences (*p* < 0.05) before and after purification.

**Table 5 foods-13-03737-t005:** IC_50_ values of α-amylase and α-glucosidase inhibition (mg/mL).

	*DO*L Extracts by VFD	*DO*L Extracts by MD	*DO*L Extracts by HAD	*DO*LP
α-amylase	5.71 ± 0.02 b	4.80 ± 0.02 a	6.68 ± 0.08 c	9.78 ± 0.09 d
α-glucosidase	7.06 ± 0.03 c	5.07 ± 0.04 a	6.40 ± 0.01 b	14.39 ± 0.21 d

Different lowercase letters indicate significant differences (*p* < 0.05) for the same enzyme.

## Data Availability

The original contributions presented in this study are included in the article, and further inquiries can be directed to the corresponding author.

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
