# Peer review of "Ultrasonic Extraction of Polysaccharides from Dendrobium officinale Leaf: Kinetics, In Vitro Activities, and Characterization"

_foods, 2024, doi:10.3390/foods13233737_

Round 1
Reviewer 1 Report
Comments and Suggestions for Authors
Optimization of extraction methods for increased extraction efficiency and development novel methods for the extraction of bioactive components have high relevance not just for the science but also for the practice. Ultrasound assisted extraction is considered as a promising method for the extraction of heat sensitive components with high efficiency. Further processing of extracts has also effect on the bioactivity. Therefore, the topic of the manuscript foods-3321668 can be considered as relevant.
The manuscript has a logic structure. Introduction summarizes well the relevance and the novelties of the study. The methods are described clearly and in details. The manuscript contains interesting and novel results that are represented well.
Comments, suggestions:
The title of the Manuscript is not suggest that the study has a special focus o the efficiency of different dehydration methods as well.
Abstract is too general. Please give the main findings, supported by with concrete data.
Please give the operating frequency of sonicator.
Please give the temperature change during sonication (power of 400 W for 30 minutes cause temperature increment).
Please give the details of different drying procedures 8frequency for MW drying, pressure/vacuum for vacuum-freeze drying etc.).
Discussion of results in section 3.1. is poorly referenced.
Please provide standard deviation for data presented in figures as well.
The visibility of figures is very poor. Please improve it (mainly axis titles, units).
Author Response
|
Response to Reviewer 1 Comments
|
||
|
1. Summary |
|
|
|
We would like to express our sincere gratitude for the reviewer's insightful comments regarding our manuscript entitled “Ultrasonic extraction of polysaccharides from Dendrobium officinale leaf : Kinetics, in vitro activities and characterization” (foods-3321668). Those comments are all valuable and very helpful for revising and improving our paper, as well as the important guiding significance to our research. We have studied comments carefully and have made correction which we hope meet with approval. Revised portion are marked in red in the paper.
|
||
|
2. Questions for General Evaluation |
Reviewer’s Evaluation |
Response and Revisions |
|
Does the introduction provide sufficient background and include all relevant references? |
Yes/Can be improved/Must be improved/Not applicable |
|
|
Are all the cited references relevant to the research? |
Yes/Can be improved/Must be improved/Not applicable |
|
|
Is the research design appropriate? |
Yes/Can be improved/Must be improved/Not applicable |
|
|
Are the methods adequately described? |
Yes/Can be improved/Must be improved/Not applicable |
|
|
Are the results clearly presented? |
Yes/Can be improved/Must be improved/Not applicable |
|
|
Are the conclusions supported by the results? |
Yes/Can be improved/Must be improved/Not applicable |
|
|
3. Point-by-point response to Comments and Suggestions for Authors |
||
|
Comments 1: The title of the manuscript is not suggest that the study has a special focus on the efficiency of different dehydration methods as well. |
||
|
Response 1: First of all, I would like to thank the reviewer for your profound insight and valuable suggestions. As demonstrated in our manuscript, various drying methods significantly influence the biological activity of Dendrobium officinale leaf extracts. Subsequent studies are conducted based on these findings, highlighting the importance of drying methods in this research, as they serve to connect previous and subsequent investigations. The reason for not including dehydration methods in the title is that we consider the extracts undergo further purification processes, and the biological activity of the purified polysaccharides is also examined. This aspect is equally important. To ensure generality and overall coherence in the title, the impact of drying methods is not explicitly stated therein. |
||
|
Comments 2: Abstract is too general. It is necessary to give the main findings, supported by with concrete data. |
||
|
Response 2: It is indeed accurate, as noted by the Reviewer, that the abstract lacks specificity. The revised manuscript illustrates our improvements to the primary findings, supported by a thorough presentation of data. (page 1, lines 14 to 16 and 21 to 25) |
||
|
Comments 3: It is necessary to provide the operating frequency of the sonicator. |
||
|
Response 3: We are very sorry for our negligence of providing the operating frequency of the sonicator. The ultrasonic processor (JY98-ⅡN, Ningbo Scientz Biotechnology Co., Ltd., Zhejiang, China) operates at a frequency of 21.00 KHz, with an ultrasonic interval of 1 second and a pause duration of 2.5 seconds. (page 3, lines 99 to 101) |
||
|
Comments 4: It is necessary to give the temperature change during sonication (power of 400 W for 30 minutes cause temperature increment). |
||
|
Response 4: In the context of temperature monitoring and control, this study employed the ice-water method to maintain a constant temperature of 0 °C during ultrasonication. Real-time temperature monitoring was conducted using a temperature probe. (page 3, lines 101 to 103) |
||
|
Comments 5: It is necessary to give the details of different drying procedures & frequency for MW drying, pressure/vacuum for vacuum-freeze drying etc. |
||
|
Response 5: We sincerely apologize for our oversight regarding the specifics of various drying procedures. The specific parameters are as follows: MD: The DOL extracts were subjected to microwave vacuum drying using a WZD2S model (Nanjing Sanle Microwave Technology Development Co., Ltd. Jiangsu, China), operating at a frequency of 2450 MHz and a power rating of 2 kW. The microwave power was set to 200 W, with each cycle lasting for 1 min followed by natural cooling for 2 min. The total duration of microwave exposure amounted to 45 min. VFD: Initially, the DOL extracts were frozen at -20 °C for a period of 24 hours. They were then transferred to a vacuum freeze-dryer (Scientz-N, Ningbo Scientz Biotechnology Co., Ltd. Zhejiang, China) where they underwent drying at an ambient temperature of 25 °C, with the cold trap maintained at -50 °C and under a vacuum pressure of 10 Pa. This drying process continued for 37 hours until a constant weight was achieved, indicating that the drying process had been successfully completed. (page 4, lines 166 to 170 and 172 to 178) |
||
|
Comments 6: Discussion of results in section 3.1. is poorly referenced. |
||
|
Response 6: It is indeed accurate, as noted by the Reviewer, that the references in section 3.1 are insufficiently cited. In response to this concern, we have added two additional references [53] and [54]. However, due to the limited availability of relevant sources on this topic, the overall number of citations remains constrained even after our revisions. (page 9, lines 358 and 377 to 379) |
||
|
Comments 7: It is necessary to provide standard deviation for data presented in figures as well. |
||
|
Response 7: We sincerely appreciate the Reviewer's suggestions. It is possible that some error bars may not be clearly visible. However, we would like to clarify that all data presented in the charts includes error bars. Additionally, to facilitate further verification by the Reviewer, we have provided the original data corresponding to the relevant charts. The Excel file has been attached for the reviewer's examination. |
||
|
Comments 8: It is essential to enhance the visibility of figures, particularly with regard to axis titles and units. |
||
|
Response 8: We sincerely appreciate reviewer's suggestions. We would like to clarify that original figures are indeed clear; however, they were compressed after being uploaded to a word document in order to reduce file size and enhance loading speed. The original figures will be provided individually at a later time. |
||
|
4. Response to Comments on the Quality of English Language |
||
|
The reviewer's comments are of high quality, characterized by clear and precise English, appropriate terminology usage, and a logical structure that is easy to comprehend. Their suggestions are specific and targeted, providing valuable insights for the enhancement of the paper. Overall, this constitutes a highly professional and detailed review that reflects the reviewer’s exceptional level of professionalism and responsible attitude. |
||
Reviewer 2 Report
Comments and Suggestions for Authors
The abstract has a correct structure and contains the necessary elements to understand the study's objective, methodology, and main findings.
The introduction section is adequate and covers the fundamental elements of the study. However, it is recommended that ultrasonication technology for extracting components, mainly polysaccharides, be explained with an emphasis. The current version is only briefly described as one tool among other options.
The manuscript's relevance is unclear, as many studies have been published where the extraction of polysaccharides from Dendrobium officinale was optimized using ultrasonication. Also, some anti-diabetic properties have been reported. Authors are advised to include information in the introduction highlighting the present manuscript's novelty.
Methodology:
In section 2.2, it is recommended that the brand and model of the ultrasonic equipment used be specified. Also, it is recommended that the ultrasonic processing be described in detail. For example, How was it set to 400 W? Was the temperature monitored and controlled? Were on/off cycles used?
The protocol and experimental design used for polysaccharide extraction need to be clarified. Initially, 400 W was used for the extraction. Later, it is mentioned that different powers (200-400 W) were used after the extraction, but the purpose of this second ultrasonic treatment needs to be clarified.
The methodology used to obtain the DOL extract needs to be clarified. Authors are advised to organize the methods and describe in detail the procedures used to facilitate understanding of the study.
Authors are encouraged to describe the statistical measures used in addition to the mean and standard deviation.
The results section and discussion are well-styled and relevant. The figures and tables are precise and well-formatted, and the statistical analysis is visible in the results. However, the methodological section must be clarified to confirm the results, mainly the experimental design, where various ultrasonication powers (W) were used after polysaccharide extraction. Furthermore, the methodology needs to describe how to obtain the DOL.
It is also recommended that citations and references be selected appropriately.
Author Response
|
Response to Reviewer 2 Comments
|
||
|
1. Summary |
|
|
|
We would like to express our sincere gratitude for the reviewer's insightful comments regarding our manuscript entitled “Ultrasonic extraction of polysaccharides from Dendrobium officinale leaf : Kinetics, in vitro activities and characterization” (foods-3321668). Those comments are all valuable and very helpful for revising and improving our paper, as well as the important guiding significance to our research. We have studied comments carefully and have made correction which we hope meet with approval. Revised portion are marked in red in the paper.
|
||
|
2. Questions for General Evaluation |
Reviewer’s Evaluation |
Response and Revisions |
|
Does the introduction provide sufficient background and include all relevant references? |
Yes/Can be improved/Must be improved/Not applicable |
|
|
Are all the cited references relevant to the research? |
Yes/Can be improved/Must be improved/Not applicable |
|
|
Is the research design appropriate? |
Yes/Can be improved/Must be improved/Not applicable |
|
|
Are the methods adequately described? |
Yes/Can be improved/Must be improved/Not applicable |
|
|
Are the results clearly presented? |
Yes/Can be improved/Must be improved/Not applicable |
|
|
Are the conclusions supported by the results? |
Yes/Can be improved/Must be improved/Not applicable |
|
|
3. Point-by-point response to Comments and Suggestions for Authors |
||
|
Comments 1: It is recommended that ultrasonication technology for extracting components, mainly polysaccharides, be explained with an emphasis. |
||
|
Response 1: As noted by the Reviewer, we recognize that we have neglected to emphasize the significance of ultrasound-assisted extraction technology, particularly in the context of polysaccharide extraction. Consequently, we have made revisions to the introduction section of the manuscript. The detailed information is as following: “Among the aforementioned methods, ultrasonication has gained significant traction in the extraction of bioactive polysaccharides due to its advantages in enhancing diffusivity, solubility, and transport of solute molecules. This method not only increases the extraction rate of polysaccharides but also reduces both the extraction time and solvent waste[28].” ( page 2, lines 65 to 69) |
||
|
Comments 2: The manuscript's relevance is unclear, as many studies have been published where the extraction of polysaccharides from Dendrobium officinale was optimized using ultrasonication. Also, some anti-diabetic properties have been reported. Authors are advised to include information in the introduction highlighting the present manuscript's novelty.” and “The relevance of the manuscript remains ambiguous, and there are shortcomings in highlighting its originality. |
||
|
Response 2: We would like to extend our sincere gratitude to the Reviewer for their invaluable advice and comments. In response, we have incorporated an additional paragraph into the introduction of the manuscript to address the issue of relevance. It is as following: “Although studies have been conducted on the optimization of ultrasound-assisted extraction of polysaccharides from the stems and leaves of DO, the optimized extraction conditions that yield a high quantity of polysaccharides were derived primarily by examining the relationship between extraction parameters and polysaccharide yield. However, the underlying principles governing polysaccharide dissolution, particularly their kinetics and thermodynamics under ultrasonic conditions, remain unclear.” Furthermore, this study primarily investigates the impact of different drying methods on the bioactivity of DOL extracts, as well as exploring the differences in bioactivity between DOL extracts and DOLP. These elements underscore the innovative contributions presented in our manuscript. (page 2, lines 71 to 77) |
||
|
Comments 3: In section 2.2, the specific brand and model of the ultrasonic equipment utilized should be clearly indicated. |
||
|
Response 3: We are very sorry for our negligence of providing the brand and model of the sonicator. The manuscript has been enhanced with additional information. The ultrasonic equipment used in this study was ultrasonic processor (JY98-ⅡN, Ningbo Scientz Biotechnology Co., Ltd. Zhejiang, China). (page 3, lines 99 to 101) |
||
|
Comments 4: The ultrasonic processing should be described in detail. |
||
|
Response 4: It is really true, as pointed out by the Reviewer, that the details regarding ultrasonic processing were not thoroughly articulated. The revised manuscript has been improved with this additional information. The specifics are as follows: “The frequency used for the ultrasonic processor (JY98-ⅡN, Ningbo Scientz Biotechnology Co., Ltd. Zhejiang, China) was 21.00 KHz, with an ultrasonic interval of 1 second and a stopping time of 2.5 seconds.” (page 3, lines 99 to 101) |
||
|
Comments 5: The protocol and experimental design used for polysaccharide extraction need to be clarified. Initially, 400 W was used for the extraction. Later, it is mentioned that different powers (200-400 W) were used after the extraction, but the purpose of this second ultrasonic treatment needs to be clarified. |
||
|
Response 5: We sincerely appreciate the Reviewer's profound insights regarding this matter. In order to determine the optimal solid-liquid ratio, we referenced parameters of 400W and 30 min from literature that is highly relevant to this study. Upon identifying the ideal material-to-liquid ratio, we proceeded to optimize both ultrasonic power and exposure time. To avoid any potential misunderstandings, we explicitly indicated in the revised manuscript that these parameters were derived from the existing literature. (page 3, lines 103 to 105) |
||
|
Comments 6: The methodology used to obtain the DOL extract needs to be clarified. |
||
|
Response 6: As suggested by the Reviewer, it is essential that we provide a comprehensive clarification of the method employed to obtain DOL extracts. The detailed methodology, which has already been incorporated into the relevant section of the revised manuscript, is as follows: “Twenty grams of DOL powder were placed in a beaker, and ultrapure water was used as the solvent, and the test was carried out at a material-liquid ratio of 1 : 40, with the ultrasonic power set at 400 W and the time set at 50 min for ultrasonic extraction. Following this, the extract was centrifuged at 4000 rpm for 15 min, and the supernatant was then concentrated by spinning at 50 °C. Subsequently, four times the volume of anhydrous ethanol was added to the concentrated extract and alcohol precipitation was carried out for 24 h. Afterward, the alcohol precipitate was centrifuged at 4000 rpm for 15 min. Finally, the precipitate, referred to as DOL extracts, underwent a dilution process and was subsequently dried utilizing three drying techniques.” (page 4, lines 157 to 165) |
||
|
Comments 7: Statistical measures, in addition to the mean and standard deviation, should be noted. |
||
|
Response 7: In response to the Reviewer′s suggestion, we have included a detailed description of the methodology employed for assessing significant differences. (page 7, lines 309 to 311) |
||
|
Comments 8: Citations and references should be selected appropriately. |
||
|
Response 8: It is indeed accurate, as pointed out by the reviewer, that citations and references are necessary for improvement. We have revised several citations and updated certain references accordingly, which includes the addition of references of [53] and [54], as well as the removal of reference [64] from the original manuscript. (page 9, lines 358 and 377 to 379) |
||
|
4. Response to Comments on the Quality of English Language |
||
|
The reviewer's comments are of high quality, characterized by clear and precise English, appropriate terminology usage, and a logical structure that is easy to comprehend. Their suggestions are specific and targeted, providing valuable insights for the enhancement of the paper. Overall, this constitutes a highly professional and detailed review that reflects the reviewer’s exceptional level of professionalism and responsible attitude. |
||